# Anomalous time delays and quantum weak measurements in optical micro-resonators

M. Asano[1,*], K.Y. Bliokh[2,3,*], Y.P. Bliokh[4,*], A.G. Kofman[2,5], R. Ikuta[1], T. Yamamoto[1], Y.S. Kivshar[3], L. Yang[6], N. Imoto[1], Ş.K. Özdemir[6] & F. Nori[2,7]

Quantum weak measurements, wavepacket shifts and optical vortices are universal wave phenomena, which originate from fine interference of multiple plane waves. These effects have attracted considerable attention in both classical and quantum wave systems. Here we report on a phenomenon that brings together all the above topics in a simple one-dimensional scalar wave system. We consider inelastic scattering of Gaussian wave packets with parameters close to a zero of the complex scattering coefficient. We demonstrate that the scattered wave packets experience anomalously large time and frequency shifts in such near-zero scattering. These shifts reveal close analogies with the Goos–Hänchen beam shifts and quantum weak measurements of the momentum in a vortex wavefunction. We verify our general theory by an optical experiment using the near-zero transmission (near-critical coupling) of Gaussian pulses propagating through a nano-fibre with a side-coupled toroidal micro-resonator. Measurements demonstrate the amplification of the time delays from the typical inverse-resonator-linewidth scale to the pulse-duration scale.

[1] Graduate School of Engineering Science, Osaka University, Toyonaka, Osaka 560-8531, Japan. [2] Quantum Condensed Matter Research Group, Center for Emergent Matter Science, RIKEN, 2-1 Hirosawa, Wako-shi, Saitama 351-0198, Japan. [3] Nonlinear Physics Centre, RSPE, The Australian National University, Canberra, Australian Capital Territory 0200, Australia. [4] Physics Department, Technion–Israel Institute of Technology, Haifa 32000, Israel. [5] Department of Chemical Physics, Weizmann Institute of Science, Rehovot 7610001, Israel. [6] Department of Electrical and Systems Enginnering, Washington University, St Louis, Missouri 63130, USA. [7] Physics Department, University of Michigan, Ann Arbor, Michigan 48109-1040, USA. * These authors contributed equally to this work. Correspondence and requests for materials should be addressed to K.Y.B. (email: k.bliokh@gmail.com) or to Ş.K.Ö. (email: ozdemir@wustl.edu).

nterference of linear waves produces many non-trivial and counter-intuitive phenomena in wave physics. Examples, which attracted considerable attention in the past two decades, include the following: optical vortices with phase singularities[1–3], curvilinear free-space propagation of Airy beams[4–6], anomalous tunnelling times and superluminal propagation of wave packets[7–10], lateral shifts of reflected or refracted beams, violating geometrical-optics rules[11–17], anomalous local group velocities and photon trajectories[18–20], and super-oscillations[21–24].

All these phenomena can appear in classical optical or microwave systems and for quantum matter waves. Moreover, anomalous shifts of quantum wave packets resulted in a new paradigm in the theory of quantum measurements, namely quantum weak measurements[25–30]. Such measurements of usual quantum observables (for example, momentum or spin) can yield rather counter-intuitive results with anomalously large 'weak values': spin 100 for spin-1/2 particles and so on. In fact, these super-shifts and super-values are direct consequences of fine interference of plane waves (Fourier components) in the wave packets corresponding to the confined quantum states.

In this work, we describe and observe a phenomenon that brings together several of the above topics in a quite simple system. Namely, we consider the resonant inelastic scattering of a one-dimensional (1D) wave packet near a zero of the complex scattering coefficient. In our proof-of-principle experiment, we deal with the transmission of an optical Gaussian pulse through a nano-fibre with a side-coupled high-$Q$ microtoroid resonator near the zero of the transmission coefficient (the so-called 'critical coupling')[31,32]. We show that in such near-zero scattering, the wave packet experiences an anomalously large time delay (either positive or negative) and also a large frequency shift. Assuming that the spectral width of the wave packet is much smaller than the linewidth of the resonance, the typical time delay is estimated as the inverse linewidth, that is, the time the pulse is trapped in the resonator[9]. For the near-zero scattering, the time delay can be enhanced to the pulse duration scale, which is demonstrated in our experiment. Similarly, the frequency shift can reach the scale of the spectral width of the pulse.

Such anomalous behaviour of the near-zero scattered pulse links the well-known phenomena of time delays and super-luminal (or subluminal) propagation[7–10] with recent studies of optical beam shifts[11–17], phase singularities[1–3,18] and the quantum weak-measurement paradigm[25–30]. Namely, the time and frequency shifts correspond to real and imaginary parts of the complex time delay, in the same manner as the spatial and angular beam shifts are described by the complex beam shift[14,16,17]. Furthermore, the complex time delay can be regarded as an anomalous weak value associated with the phase singularity of the scattering coefficient. Importantly, the previously known formulas for time delays diverge in the singular zero-scattering point. Using the extended theory of quantum weak measurements[14,29], we derive simple expressions that accurately describe the anomalous (but finite) time and frequency shifts for near-zero scattering.

It should be noticed that some of the links between the above topics have been considered before, in particular the relations between beam shifts and quantum weak measurements[12,14–16], time delays and weak measurements[33–36], Goos–Hänchen beam shifts and time delays[37,38], as well as the considerable role of phase singularities in anomalous weak values[18,39]. However, the results of our work unify all these phenomena in a fairly complete way in a simple one-channel scattering problem. Most importantly, in contrast to previous studies, the phase singularity and complex weak value appear in our problem in a 1D system without internal degrees of freedom (polarization or spin). For example, a related study by Solli et al.[39] has emphasized

the connection between anomalous time delays (but not frequency shifts), phase singularities of the transmission coefficient and quantum weak measurements. However, that study essentially involved a two-dimensional (2D) microwave system with polarization degrees of freedom. Moreover, their time-delay expressions were still divergent in the zero-transmission point. In our case, a rich and non-trivial physical picture with vortices and weak values naturally arises in a genuine 1D scalar system because of its non-Hermitian character involving complex frequencies and phases.

We verified our theoretical predictions and measured anomalous time delays in experiments performed using a cutting-edge optical setup. Namely, we used 17-nanosecond Gaussian pulses propagating in a nano-fibre coupled to a high-$Q$ whispering-gallery-mode toroidal micro-resonator ($Q_0 \simeq 2.9 \cdot 10^6$). Recently, it was demonstrated that such micro-resonators are capable of revealing a number of fundamental non-Hermitian phenomena of wave physics[40–43]. In our case, the critical coupling with the resonator resulted in both positive and negative time delays (that is, subluminal and superluminal propagation) up to 15 ns.

## Results

**Resonator and time delays of scattered wave packets.** We start with the description of basic features of resonant inelastic one-channel scattering of a Gaussian wave packet. To be explicit, we consider a 1D problem with an optical pulse propagating in a waveguide (nano-fibre in our experiment) and interacting with a side-coupled high-$Q$ ring resonator (Fig. 1). Near resonance, the transmission of a single harmonic wave with angular frequency $\omega$ through the system can be described by the following transmission coefficient[31]:

$$T(\omega, \Gamma) = \frac{(\omega - \omega_0) - i(\Gamma - \Gamma_0)}{(\omega - \omega_0) + i(\Gamma + \Gamma_0)}. \qquad (1)$$

Here, $\omega_0$ is the resonant frequency of the resonator, $\Gamma_0 \ll \omega_0$ is the internal dissipation rate of the resonator and $\Gamma \ll \omega_0$ is the coupling rate between the incident wave and the resonator. It is noteworthy that in a different geometry, when a standard resonator cavity directly couples to the incoming and outgoing waveguides, the reflection coefficient has the form of equation (1)[31,32]. Therefore, all the conclusions of this work are equally applicable to the wave reflection in such geometry.

We regard the wave frequency $\omega$ and the coupling parameter $\Gamma$ as variables in equation (1), because these parameters are varied in our experiment. The $Q$-factors of the uncoupled and wave-coupled resonator are given by $Q_0 = \omega_0/2\Gamma_0 \gg 1$ and $Q = \omega_0/2(\Gamma_0 + \Gamma) \gg 1$, respectively; the latter one determines the linewidth of the resonant transmission. Note that equation (1) describes the wave transmission in the vicinity of the resonance line, that is, when $|\omega - \omega_0| \lesssim (\Gamma_0 + \Gamma) \ll \omega_0$, and it is not valid for $|\omega - \omega_0| \gg (\Gamma_0 + \Gamma)$.

Equation (1) has a universal form, which can be regarded as the generalized Breit–Wigner formula for the S-matrix of a one-channel resonant scattering in quantum mechanics[9,44]. However, instead of poles of the scattering matrix, which are usually considered in scattering theory, we are interested here in zeros of the transmission coefficient (1). Namely, when the wave frequency matches the resonator frequency, $\omega = \omega_0$, and the coupling coefficient coincides with the internal dissipation in the resonator, $\Gamma = \Gamma_0$, the so-called critical coupling takes place[31,32]. Under these conditions, the transmission vanishes, $T(\omega_0, \Gamma_0) = 0$, and all the wave energy is absorbed by the resonator. Near the critical-coupling parameters, the transmission coefficient behaves similar to a generic complex function near its zero, that is, forms a

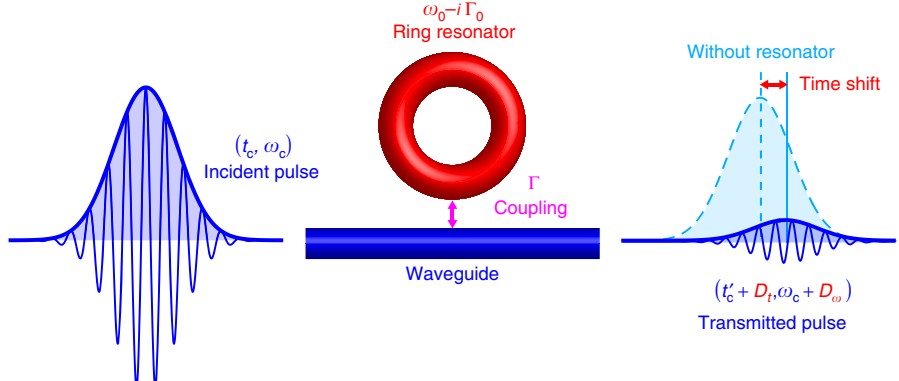

**Figure 1 | Time and frequency shifts of an optical pulse interacting with a waveguide-coupled resonator.** An incident Gaussian wave packet with central frequency $\omega_c$ and intensity-maximum time $t_c$ in the starting point propagates through a waveguide with a side-coupled ring resonator. The resonator is characterized by a resonant frequency $\omega_0$, dissipation rate $\Gamma_0$, whereas the coupling rate between the resonator and the waveguide is denoted by $\Gamma$. In the absence of the resonator, the intensity of the transmitted pulse is expected to be maximum at the time $t'_c$ in the point of observation. Interacting with the resonator, the transmitted pulse experiences shifts in both its arrival time (with time delay $D_t$, shown negative here) and its central frequency (frequency shift $D_\omega$). These shifts are strongly enhanced near the critical-coupling (zero-scattering) regime, when most of the pulse energy is absorbed by the resonator and the transmitted-pulse amplitude is small.

vortex with a phase singularity[1–3,39]. Indeed, introducing complex detuning from the critical-coupling parameters $\bar{v} = (\omega - \omega_0) - i(\Gamma - \Gamma_0) \equiv v - i\gamma$, equation (1) behaves as $T(\bar{v}) \simeq -i\bar{v}/2\Gamma_0$ for $|\bar{v}| \ll \Gamma_0$.

We now consider a Gaussian wave packet or pulse consisting of multiple waves with different frequencies. The field of the incident Gaussian wave packet can be written in the frequency and time representations as:

$$\tilde{E}(\omega) \propto \exp\left[-\frac{(\omega - \omega_c)^2}{2\tilde{\Delta}^2}\right], \quad E(t) \propto \exp\left[-i\omega_c t - \frac{(t - t_c)^2}{2\Delta^2}\right]. \quad (2)$$

Here, $\omega_c$ is the central frequency of the packet, $\tilde{\Delta}$ is the spectral width of the pulse and $\Delta = 1/\tilde{\Delta}$ is the temporal length of the pulse. In equation (2), we consider temporal variations of the wavepacket field in the point of observation (say, $x = 0$), assuming that the field amplitude is maximal at $t = t_c$.

We also assume that the central frequency of the wave packet is close to the resonant frequency of the resonator, so that equation (1) is applicable for $\omega = \omega_c$, and that the spectral width of the wave packet is much smaller than the linewidth of the resonance (1). These conditions can be written as

$$|\omega_c - \omega_0| \leq (\Gamma_0 + \Gamma), \quad \tilde{\Delta} \ll (\Gamma_0 + \Gamma). \quad (3)$$

The second condition (3) is the 'weak-coupling' or 'adiabatic' condition, which implies that the Gaussian shape of the wave packet is only weakly perturbed by the interaction with the resonator (apart from the overall scaling). Assuming that $\Gamma \sim \Gamma_0$, we will use the small weak-coupling (adiabatic) parameter $\varepsilon = \tilde{\Delta}/\Gamma_0 \ll 1$.

In the zero-order approximation in $\varepsilon$, the field of the transmitted pulse is given by $\tilde{E}'(\omega) \simeq T(\omega_c)\tilde{E}(\omega)$. As the transmitted pulse is observed at some point $x = L$, its temporal form is $E'(t) \simeq T(\omega_c)E(t')$, where $t' \to t - L/c$, with $c$ being the (group) velocity of the wave in the waveguide. Thus, the field of the transmitted pulse is expected to be maximal at the time $t'_c = t_c + L/c$ in the point of observation (see Fig. 1).

Taking into account the finite spectral width of the pulse and different complex transmission coefficients for waves with different frequencies, one can see that the transmitted pulse is perturbed by interesting interference phenomena. In the first-order approximation in $\varepsilon$, we can use the Taylor expansion

of the transmission coefficient near the central frequency: $T(\omega) \simeq T(\omega_c) + (\partial T(\omega_c)/\partial\omega_c)(\omega - \omega_c)$. Then, the Fourier spectrum of the transmitted pulse becomes:

$$\tilde{E}'(\omega) \simeq T(\omega_c)\left[1 + \frac{\partial \ln T(\omega_c)}{\partial \omega_c}(\omega - \omega_c)\right]\tilde{E}(\omega). \quad (4)$$

The second term in square brackets in equation (4) originates from the dispersion of the transmission coefficient. It contains the frequency $\omega$ and therefore affects the shape of the transmitted pulse in the time representation (where frequency becomes the operator $\hat{\omega} = i\partial/\partial t$).

Using precise analogy of the transformation (4) with the analogous spatial transformation in the optical beam-shift and quantum weak-measurements problems[16,17,29] (which is described below), one can show that the transmitted pulse acquires the complex time delay $D$:

$$E'(t) \simeq T(\omega_c)E(t' - D), \quad D = -i\frac{\partial \ln T(\omega_c)}{\partial \omega_c}. \quad (5)$$

In terms of real-valued quantities, the transmitted field can be presented in Gaussian form in both frequency and time domains:

$$E'(t) \simeq T(\omega_c)E(t' - D_t)e^{-iD_\omega(t - t_c)}, \quad D_t = \mathrm{Re}D, \quad (6)$$

$$\tilde{E}'(\omega) \simeq T(\omega_c)\tilde{E}(\omega - D_\omega)e^{iD_t(\omega - \omega_c)}, \quad D_\omega = -\tilde{\Delta}^2\mathrm{Im}D. \quad (7)$$

Here, $D_t$ is the well-known Wigner time delay[7–10,45], that is, a shift of the Gaussian envelope in time (and longitudinal coordinate), whereas $D_\omega$ is a small frequency shift associated with the imaginary part of the complex shift (5) (see Fig. 1). Although complex time shifts (5) were widely discussed in the literature (see refs 7–9 and references therein), it was not properly recognized that the imaginary part of this time is responsible for the frequency rather than time shift.

Thus, because of the interaction with the resonator and associated interference effects, the transmitted pulse is slightly shifted in both time and frequency domains with respect to the propagation without resonator. In quantum-mechanical terms, the expectation values of the arrival time and frequency (energy) of the transmitted pulse are $\langle t \rangle = t'_c + D_t$ and $\langle \omega \rangle = \omega_c + D_\omega$, respectively. Although the frequency shift looks similar to a second-order effect in $\varepsilon$, $D_\omega \propto \tilde{\Delta}^2$, it originates from the first-order complex time delay (5). Taking into account the true second-order terms in the Taylor expansion of the transmission

coefficient does not contribute to the frequency shift in this approximation. It is also noteworthy that the frequency shift does not affect the pulse propagation in non-dispersive waveguides, that is, when the group velocity $c$ is independent of $\omega$. In the dispersive case, $c = c(\omega)$, the frequency shift will modify the propagation time $t'_c$ and cause an additional time delay $D_t^{\text{dispers}} = -(L/c^2)(\partial c/\partial \omega) D_\omega$ growing with the propagation distance $L$.

Remarkably, equations (4)–(7) are precise temporal analogues of the equations for the Goos–Hänchen beam shifts, which occur in the wave-beam reflection or refraction at an optical interface[15–17]. In this manner the real part of the complex time shift (5) (that is, the Wigner time-delay formula) is an analogue of the Artmann formula[46], whereas the time and frequency shifts (6) and (7) are the counterparts of the spatial (coordinate) and angular (wave vector) Goos–Hänchen shifts[15–17]. The close analogy between the Goos–Hänchen and time-delay effects was previously recognized in refs 37,38. Notably, the imaginary part of the complex time delay was measured as the angular Goos–Hänchen shift in ref. 38, but still it was not recognized as the frequency shift. Lateral beam shifts at optical interfaces have recently attracted enormous attention in connection with spin–orbit interactions of light and quantum weak measurements[11–17]. Such shifts are studied in 2D or three-dimensioanl (3D) geometries, and they are strongly dependent on the internal polarization (spin) degrees of freedom. In contrast, the problem we deal with here involves purely scalar 1D waves, with their complex phases being the only internal degree of freedom.

The Wigner time delay $D_t$ can be either positive or negative, resulting in the effective 'subluminal' or 'superluminal' propagation of the pulse[7–10], that is, 'slow' or 'fast' light[39]. Similarly, the frequency shift $D_\omega$ can be either positive or negative. In the former case, the normalized energy 'per photon' in the transmitted pulse will be higher than that in the incident pulse. This does not violate energy conservation, because the transmitted pulse contains less number of photons (intensity) than the incident one.

Importantly, the shifts (5)–(7) diverge in the critical-coupling regime: $D \to \infty$ at $T(\omega_c) = 0$. This means that the typical time-delay values can be significantly enhanced for the parameters close to the zero of the scattering coefficient, and that the above simple equations are not applicable for the near-zero scattering. Below we show that the formalism of quantum weak

measurements perfectly describes this phenomenon and provides laconic expressions for the enhanced time and frequency shifts in the near-zero scattering regime.

**Quantum weak measurements in near-zero scattering.** The paradigm of 'quantum weak measurements' was introduced by Aharonov *et al.*[25]. Since then, numerous studies suggested various examples and interpretations of this concept[12,14,16–20,26–30,33–36,39]. Although the usual 'strong' quantum measurements result in expectation values of the corresponding operators, weak measurements bring about so-called 'weak values' of the measured quantities. Remarkably, weak values can be complex and even their real parts can be anomalously large, that is, lie outside of the spectrum of the operator. This is closely related to the phenomenon of 'superoscillations'[21–24], when the phase of a complex function varies with anomalous gradients, which are much higher than any spatial Fourier components in its spectrum.

Anomalous weak values and superoscillations are often related to vortices, that is, phase singularities or zeros of complex functions[1–3]. One of the simplest examples, proposed by Berry[18], is the measurement of the local momentum of a wave field near a vortex. Consider 2D space $\mathbf{r} = (x, y)$ and the wave function $\psi(\mathbf{r})$ with vortex at the origin, $\psi(\mathbf{0}) = 0$ (Fig. 2a). In the vicinity of this zero, the wave function behaves as $\psi(\mathbf{r}) \propto [x + i\,\text{sgn}(\ell)\, y]^{|\ell|}$, where $\ell$ is the vortex strength, which is a non-zero integer number. Weak measurements of the momentum $\hat{\mathbf{p}} = -i\partial/\partial\mathbf{r}$ conjugated to $\mathbf{r}$ (we use units $\hbar = 1$), for the state $|\psi\rangle$ with the postselection in the coordinate eigenstate $|\mathbf{r}\rangle$, result in the following weak value of the momentum[18–20]:

$$\mathbf{p}_w = \frac{\langle\mathbf{r}|\hat{\mathbf{p}}|\psi\rangle}{\langle\mathbf{r}|\psi\rangle} = -i\frac{\partial \ln\psi}{\partial\mathbf{r}}. \tag{8}$$

This 'weak momentum' is complex and it diverges in the vortex point: for example, $\text{Re}\,\mathbf{p}_w \to \infty$ at $\mathbf{r} \to 0$, $\psi(\mathbf{r}) \to 0$ (Fig. 2a). This is because the phase gradient of the wave function is anomalously high near the vortex (superoscillations). The real part of the weak value (8) represents the normalized momentum density $\text{Re}\,\mathbf{p}_w = \mathbf{p}(\mathbf{r})$ of the wave field and it is directly observable in experiments with local probes interacting with the wave field at a given point $\mathbf{r}$[20,47]. Therefore, a probe (for example, a nanoparticle or an atom immersed in an optical field $\psi(\mathbf{r})$) experience anomalous momentum transfer ('super-kicks') proportional to

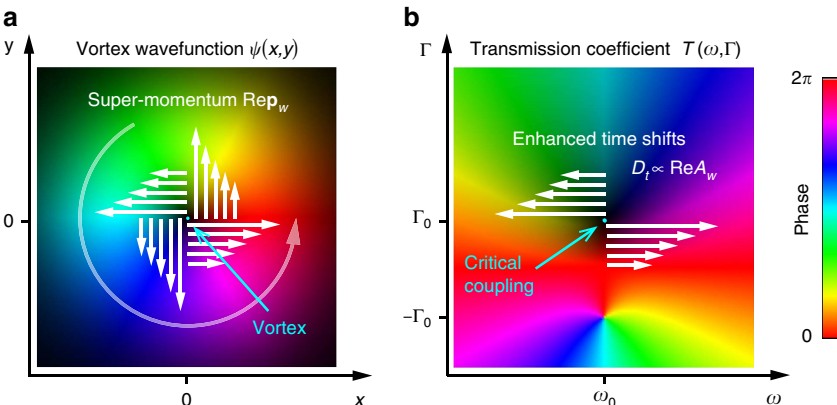

**Figure 2 | Analogy between anomalous weak momentum in a vortex wavefunction and time delays of a wave packet in near-zero scattering.** (**a**) Weak measurements of the momentum in a vortex wave field, $\psi(\mathbf{r})$, described by equation (8), result in the anomalously large weak values (super-momentum) $\mathbf{p}_w$ near the vortex core[18]. Here, the localized vortex wavefunction $\psi = (x + iy)\exp(-x^2 - y^2)$ is shown. (**b**) Anomalously large time delays $D$, given by equations (5)–(7), which appear in the vicinity of the zero of the transmission coefficient $T(\omega, \Gamma)$ (critical coupling), have the weak-value form (10) similar to equation (8). In both panels, colours indicate the phases of the complex functions, whereas brightness corresponds to their absolute values.

Re$\mathbf{p}_w$ in the vicinity of the vortex. The anomalously high value of such kicks is compensated by a very low probability of their occurrence, because the amplitude of the wave function vanishes in the vortex.

Equation (5) for the complex time delay $D$ closely resembles the weak-momentum equation (8). In our case, the complex transmission coefficient $T(\omega, \Gamma)$ plays the role of the 'wave function', where the critical-coupling point $(\omega, \Gamma) = (\omega_0, \Gamma_0)$ corresponds to a vortex of strength $\ell = -1$ (Fig. 2b). As a result, the pulse (which plays the role of the probe here) experiences a 'super-kick' in its time variable $t$ conjugated to $\omega$. The only difference with the above vortex example is that in our case we deal with a 1D system and the 2D vortex in the transmission coefficient appears because we deal with a non-Hermitian system and complex frequencies $\bar{\omega} = \omega - i\Gamma$, corresponding to this single dimension.

The above analogy between quantum weak measurements and enhanced complex pulse delay (5) can be formalized using the approach suggested by Solli et al.[39] Namely, one can write equation (4) for the pulse transmission in the form of the weak-measurement evolution equation:

$$E'(t) \propto T(\omega_c)\left[1 + iA_w\hat{F}\right]E(t). \qquad (9)$$

Here, the pulse plays the role of the probe ('metre') with variable $\hat{F} = \hat{\omega} - \omega_c$, which measures the weak value of some operator $\hat{A}$. Without knowing the actual form of the operator $\hat{A}$, its weak value is given by

$$A_w = -i\frac{\partial \ln T(\omega_c)}{\partial \omega_c} \equiv D. \qquad (10)$$

According to the general weak-measurement formalism[29,30], the imaginary and real parts of the weak value (10) produce shifts (7)

and (6) in the variable $\hat{F}$ (that is, frequency) and the variable conjugated to $\hat{F}$ (that is, time). Thus, the complex time delay (5) perfectly matches the weak-measurement paradigm as the weak value (10). Such one-to-one correspondence between the wavepacket shifts and quantum weak values was previously emphasized for Goos–Hänchen and Imbert–Fedorov (spin-Hall effect) beam shifts in the optical reflection and refraction problems[16,17].

We can now use this correspondence to regularize the singularity of the time delay in the critical-coupling regime. The time and frequency shifts (6) and (7) appear only in the linear-response regime, which assumes that the envelope of the transmitted wave packet still has a Gaussian profile[29]. However, the shape of the wave packet is strongly deformed in the vicinity of the zero of the transmission coefficient, which acts as a spectral filter, and the transmitted pulse is not Gaussian anymore[26-30]. The weak-measurements formalism allows us to obtain general expressions for the wavepacket shifts, which remain finite even when the weak value diverges (see refs 14,29):

$$D_t = \frac{\mathrm{Re}A_w}{1 + \tilde{\Delta}^2|A_w|^2/2}, \qquad D_\omega = -\frac{\tilde{\Delta}^2\mathrm{Im}A_w}{1 + \tilde{\Delta}^2|A_w|^2/2}. \qquad (11)$$

These are the main equations, which describe the anomalous time and frequency shifts of a wave packet in the near-zero scattering regime. It is noteworthy that $D_t = D_\omega = 0$ for the exact critical-coupling parameters when $T(\omega_c) = 0$, $|A_w| = \infty$.

Substituting the transmission coefficient (1) into (10), we obtain the explicit form of the weak value (complex time delay):

$$A_w = \frac{2\Gamma}{(\omega_c - \omega_0)^2 + (\Gamma^2 - \Gamma_0^2) + 2i\Gamma_0(\omega_c - \omega_0)}. \qquad (12)$$

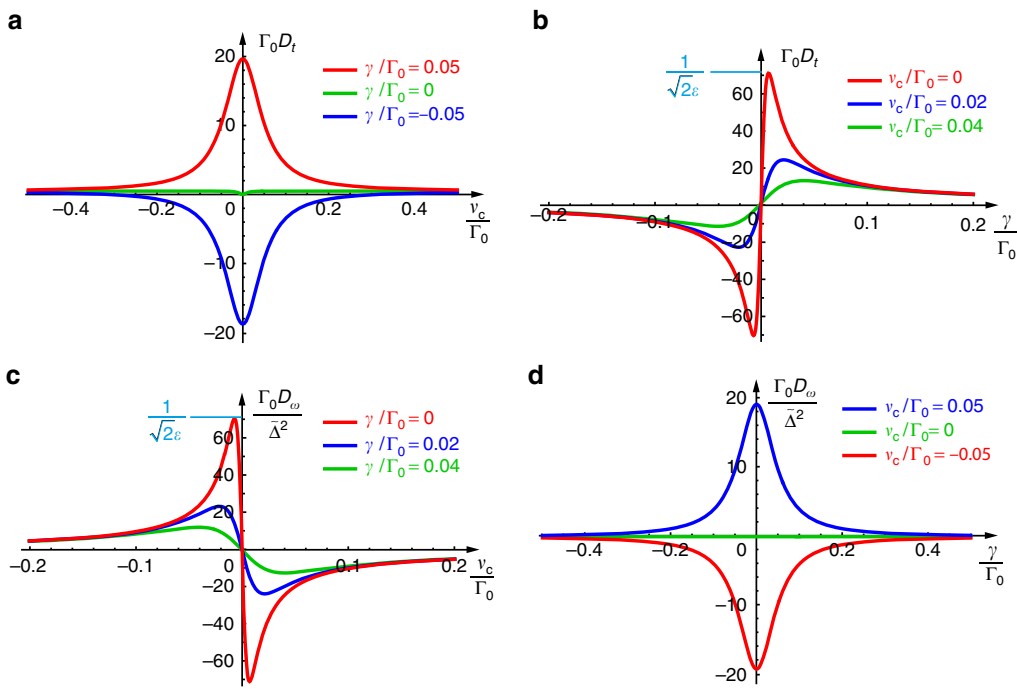

**Figure 3 | Theoretically calculated time and frequency shifts of an optical pulse transmitted through a waveguide-coupled resonator.** Time (**a,b**) and frequency (**c,d**) shifts of the transmitted pulse, $D_t$ and $D_\omega$, described by the weak measurement equations (11) and (12), versus frequency and coupling detunings, $v_c = \omega_c - \omega_0$ and $\gamma = \Gamma - \Gamma_0$. The shifts are strongly enhanced near the critical-coupling (zero-transmission) point $(v_c, \gamma) = (0,0)$. The adiabatic parameter is taken here as $\varepsilon = 0.01$. The normalization constants are chosen in such a way that the dimensionless shift values indicate their enhancements over the typical shifts (without critical coupling). The extreme values of the dimensionless shifts ($\simeq \pm 1/\sqrt{2}\varepsilon$), given by equations (13) and (15), are seen in the red curves in **b,c**.

From here and equations (11), we derive that extreme time delays are achieved at (i) the resonant frequency of the pulse and (ii) for the coupling slightly shifted from the critical value:

$$D_{t\,\max} = \pm \frac{1}{\sqrt{2}\tilde{\Delta}} \quad \text{for} \quad v_c = \omega_c - \omega_0 = 0,$$

$$\gamma = \Gamma - \Gamma_0 \simeq \pm \frac{\tilde{\Delta}}{\sqrt{2}}. \tag{13}$$

As the typical Wigner time delay away from the critical-coupling region can be estimated as $|D_t| \sim 1/\Gamma_0$, the maximal weak-measurement amplification of the time delay is given by the factor:

$$\Lambda = \frac{\Gamma_0}{\tilde{\Delta}} = \frac{1}{\varepsilon} \gg 1. \tag{14}$$

In other words, for the near-zero scattering, the time delays can be amplified from the inverse resonator linewidth scale to the pulse-length scale.

In a similar manner, the frequency shift reaches its extreme values at the critical value of the coupling and slight detuning of the central frequency of the pulse:

$$D_{\omega\,\max} \simeq \pm \frac{\tilde{\Delta}}{\sqrt{2}} \quad \text{for} \quad v_c \simeq \pm \frac{\tilde{\Delta}}{\sqrt{2}}, \quad \gamma = 0. \tag{15}$$

Thus, the frequency shift can achieve values of the order of the spectral pulse width.

Figure 3 shows plots of the time and frequency shifts (11) and (12) versus frequency and coupling detunings from their critical-coupling values. These curves have a Lorentzian and resonant shapes typical for quantum weak-measurement problems[14,29,30,48]. It is worth noting that the dependences $D_t(v_c)$ and $D_\omega(\gamma)$ are similar to each other in shape, as well as the $D_t(\gamma)$ and $D_\omega(v_c)$ dependences.

**Experimental results**. To test the above theoretical predictions, we performed an experiment involving the transmission of an optical pulse through a nano-fibre with a side-coupled whispering-gallery-mode toroidal micro-resonator.

Figure 4 shows schematics of the experimental setup. The silica micro-toroid resonator on a silicon chip was fabricated by photolithography followed by isotropic etching of silicon with

xenon difluoride and $CO_2$ laser re-flow. For the measurements of time delay of optical pulses, a tunable external cavity diode laser (ECDL) was modulated with an electro-optic modulator (EOM) by a burst sine-shaped electric signal sent from an arbitrary function generator. A tapered nano-fibre prepared from a standard single-mode optical fibre by heat-and-pull technique was used to couple light to the micro-resonator after adjusting an appropriate polarization and power of light by a fibre-based polarization controller and an attenuator, respectively. The transmitted optical pulses were detected using a photodetector (PD) connected to a digital sampling oscilloscope.

To determine time shifts of the transmitted pulses, a reference pulse was initially measured in the setup without the resonator. The temporal data were simultaneously collected (ten times per single measurement with fixed parameters) by the digital sampling oscilloscope synchronized to the EOM using a digital delay generator at 100 kHz. The intrinsic quality factor of the resonator, $Q_0 = \omega_0/2\Gamma_0$, was measured from the half-maximum width of the transmission spectrum by sweeping the frequency of the ECDL. This yielded $Q_0 \simeq 2.9 \cdot 10^6$ for the resonance, which was used for the following measurements.

We controlled the two main parameters in the experiment: the laser detuning from the resonance frequency of the resonator, $v_c = \omega_c - \omega_0$ and the coupling strength $\Gamma$ between the resonator and the nano-fibre.

First, the detuning $v_c$ was adjusted by a feedback system, which consists of: a fibre-based Mach–Zender interferometer (FMZI) immersed in water, to remove the mechanical fluctuation from the environment, a balanced amplified PD (BAPD) and a PC and a voltage controller. To obtain an error signal, we pick up a part of the continuous wave light emitted by the ECDL before modulating with the EOM by 1:99 beam splitter and send it to the FMZI. The information of detuning was obtained from the dual outputs of the FMZI, which were measured by a BAPD[49]. The difference signal (that is, electric error signal) generated in the BAPD was used to calculate the feedback voltage. This voltage was then generated in the voltage controller and sent to the piezoelectric transducer of the ECDL that controls the position of the grating and hence the laser frequency of the ECDL.

Second, the coupling strength $\Gamma$ was controlled by varying the gap between the fibre and the resonator using an open-loop 3D nano-positioning system. The actual varying parameter was the

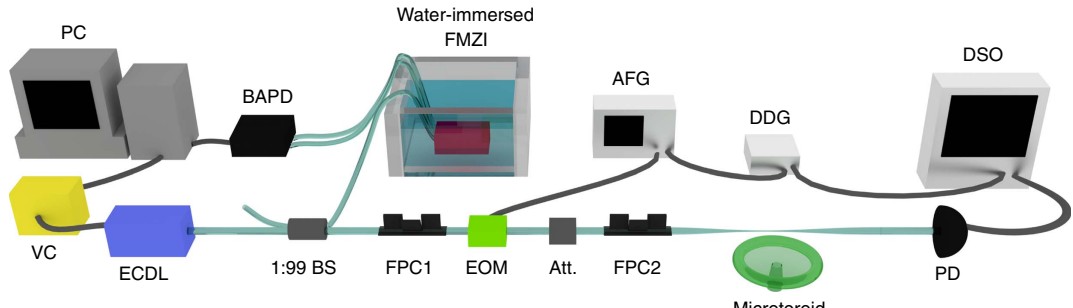

**Figure 4 | Schematics of the experimental setup.** The light pulses are prepared by modulating the light from a tunable ECDL using an EOM driven by an arbitrary function generator (AFG). The prepared pulses are coupled in and out of the silica microtoroid resonator using a tapered optical fibre. The transmitted light is detected by a PD connected to a digital sampling oscilloscope (DSO). The DSO and the EOM are synchronized with the digital delay generator (DDG). The coupling is adjusted by controlling the distance between the tapered fibre and microtoroid. The required frequency detuning for the experiments is achieved with the help of a FMZI. For this purpose, a portion (~1%) of the ECDL output is tapped out with a 1:99 beamsplitter (BS) and sent to the input port of the FMZI. The output of the FMZI is detected by a BAPD, which produces an error signal when the frequency of the ECDL deviates from the pre-determined value. The FMZI is immersed in a water environment to reduce vibrations and to block any ambient air changes. The generated error signal is processed and a suitable control signal is generated using a voltage controller (VC). The control signal is applied to the piezoelectric transducer of the ECDL to control the frequency of the light emitted by the ECDL. The polarization of the light before and after the EOM is adjusted using fibre-polarization controllers (FPC1 and FPC2). An attenuator (Att.) is used to attenuate the light before it is coupled to the resonator to prevent thermal and nonlinear effects.

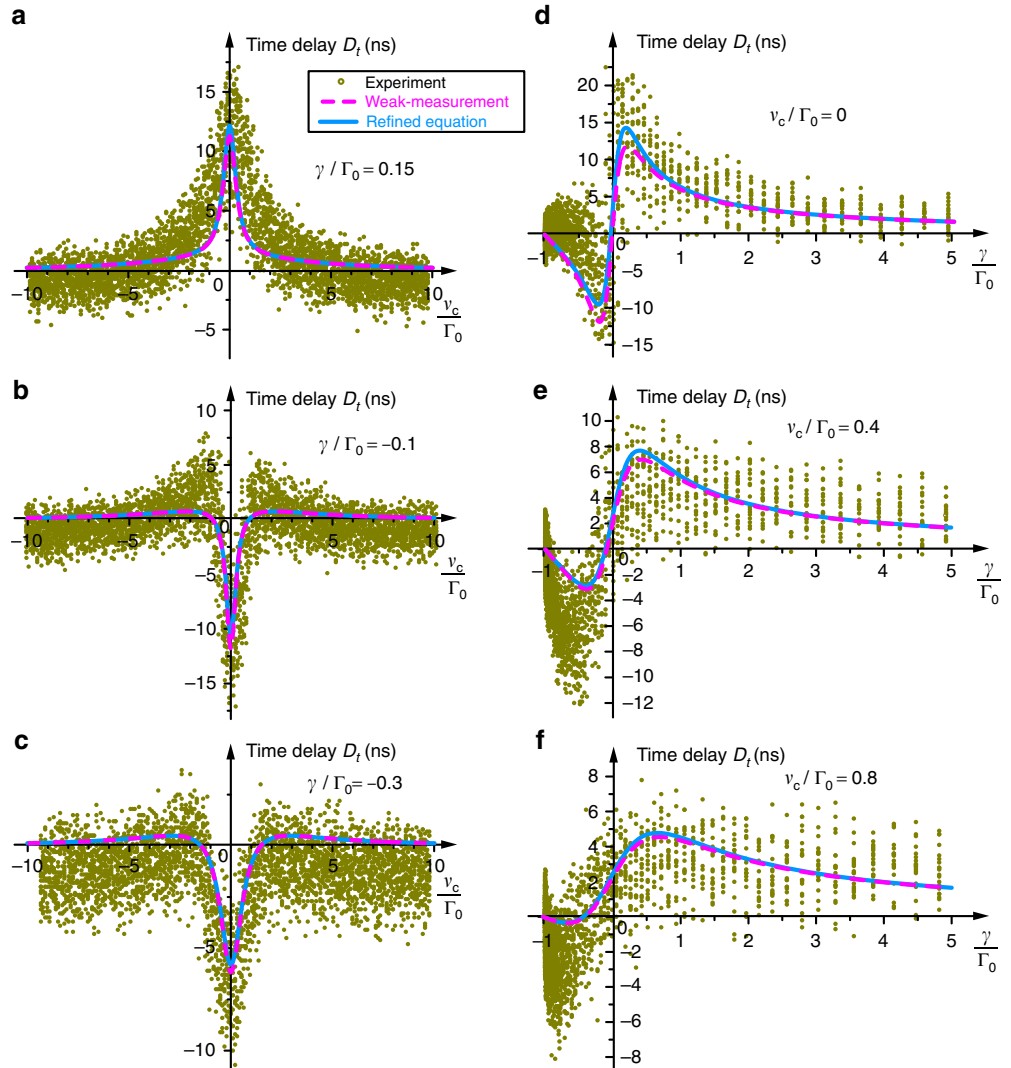

**Figure 5 | Experimentally measured anomalous time shifts of light pulses transmitted through a waveguide-coupled resonator.** Time delays $D_t$ of the transmitted pulses as functions of (**a**–**c**) the frequency detuning $v_c$ at different coupling parameters $\gamma$ and (**d**–**f**) the coupling detuning $\gamma$ at different frequency detunings $v_c$. Each symbol corresponds to a single time-delay measurement. The magenta dashed curves represent the theoretical weak measurement equations (11) and (12). The cyan solid curves represent the theoretical results obtained using the refined equations, which include the second derivative of the transmission coefficient (see Supplementary Note 2). Despite the large dispersion of the experimental data, the resonant behaviour in the vicinity of the critical coupling $(v_c, \gamma) = (0, 0)$ is clearly seen and the behaviour of the time delays is in good agreement with the theoretical predictions.

voltage $V$ of the nano-positioner. It varied the distance $d$ between the fibre and the resonator: $d \propto V$. The coupling between the fibre and the resonator is realized via evanescent fields, which decay exponentially with $d$. Therefore, the coupling strength is related to the positioner voltage as $\Gamma = \alpha \exp(-\beta V)$, where $\alpha$ and $\beta$ are unknown constants to be determined from the experiment.

We performed two series of experiments. In the first one, the detuning of the pulses, $v_c$, was varied in a relatively broad range, whereas the positioner voltage $V$ (and the coupling $\Gamma$) was fixed. Then, the intensity of the transmitted pulse, $|E'(t)|^2$, was measured and processed for every value of the detuning $v_c$. Calculating the time shift of the centroid (that is, 'centre of gravity' of the intensity distribution) of the transmitted pulse with respect to the reference arrival time without the resonator, we determined the experimental values of the time shift $D_t$ versus the frequency detuning $v_c$ (cf. Fig. 3a). This series of measurements was repeated for different values of the voltage $V$ (coupling $\Gamma$).

In the second series of experiments, we varied the positioner voltage $V$ at a fixed detuning $v_c$. The experimentally measured time delays $D_t$ versus the voltage $V$ showed two well-pronounced extrema (see Supplementary Fig. 1), similar to those in the theoretical curves $D_t(\Gamma)$, equations (11), (12) and Fig. 3b. Now, associating the voltages $V_{min}$ and $V_{max}$, corresponding to the extrema of the $D_t(V)$ curves, with the values $\Gamma_{min}$ and $\Gamma_{max}$, corresponding to the extrema in the theoretical dependences $D_t(\Gamma)$, we retrieved the two unknown parameters $\alpha$ and $\beta$ relating the voltage to the coupling constant. Finally, using the equation $\Gamma = \alpha \exp(-\beta V)$, we plotted the experimentally measured time delay $D_t$ versus the coupling strength $\Gamma$ or its dimensionless detuning $\gamma/\Gamma_0 = (\Gamma - \Gamma_0)/\Gamma_0$ (see Supplementary Fig. 1 and Supplementary Note 1). This series of measurements was repeated for different detunings $v_c$. Importantly, determining the constants $\alpha$ and $\beta$ from different series of measurements with different detunings $v_c$ resulted in approximately the same values (with variations $\sim 10\%$). Therefore, we calculated the

averaged values $\bar{\alpha}$ and $\bar{\beta}$ from all these series of measurements and used these values for the global mapping $\Gamma(V)$ in all the experimental data.

The results of experimental measurements of time delays $D_t(v_c, \gamma)$ and the corresponding theoretical curves are shown in Fig. 5. For every pair of parameters, we measured time shifts of $\sim 15$–$20$ pulses and all these measurements are shown in Fig. 5 as symbols. Although the dispersion of the experimental data is large, one can clearly see the resonant behaviour and the enhancement of the time delay in the vicinity of the critical-coupling regime $(v_c, \gamma) = (0, 0)$, exactly as predicted theoretically by equations (11), (12) and Fig. 3.

Importantly, the adiabatic parameter was not too small in our experiment due to technical restrictions. Namely, the resonant frequency was $\omega_0 \simeq 1.2 \cdot 10^{15} \mathrm{rad}\,\mathrm{s}^{-1}$ and the $Q_0$-factor of the resonator corresponded to the dissipation rate $\Gamma_0 \simeq 2.07 \cdot 10^8 \mathrm{s}^{-1}$. At the same time, the longest pulse we could generate in our system had the duration $\Delta \simeq 16.76 \cdot 10^{-9} \mathrm{s}$. This yields the adiabatic parameter $\varepsilon = (\Delta\,\Gamma_0)^{-1} \simeq 0.29$. Thus, our parameters correspond to the boundary of the applicability of adiabatic weak-measurement theory and one should not expect perfect qualitative agreement between the measurements and theoretical equations. Nonetheless, we clearly observe all details of the predicted time-delay behaviour. In particular, the maximal enhancement of the time delay near the critical-coupling regime was $\Lambda \sim 3.5$, in agreement with equation (14). As predicted in equation (13), the maximal time delay was of the order of the pulse duration, that is, $|D_{t\,\mathrm{max}}| \sim 12 \cdot 10^{-9} \mathrm{s}$ (Fig. 5).

As the adiabatic parameter was not too small in our experiment, we preformed additional calculations of time shifts, which take into account the second-derivative terms in the Taylor expansion of the transmission coefficient $T(\omega)$. These calculations are presented in the Supplementary Note 2 and the results are similar to the analogous beam-shift calculations by Götte and Dennis[48]. The refined dependences $D_t(v_c, \gamma)$ are plotted in Fig. 5. One can see that the curves described by the simplest weak-measurement equations (11) and (12) are still quite close to the refined curves for $\varepsilon \simeq 0.29$ although little quantitative difference is noticeable. However, basically, the adiabatic weak-measurement approximation works very well even for the given $\varepsilon$ and one can safely use equations (11) and (12).

## Discussion
We have revealed interesting peculiarities of inelastic resonant scattering of a 1D wave packet in the vicinity of a zero of the scattering coefficient. Such near-zero scattering exhibits remarkable analogy with quantum weak measurements of the momentum variable near a phase singularity of the complex wave function. In the scattering problem, this analogy manifests itself as an anomalously large time delay and frequency shift of the scattered wave packet. These are the results of fine interference of Fourier components with small amplitudes in the scattered wave packet.

The typical Wigner time delay is estimated as the inverse linewidth of the resonance, that is, the time of the wave packet trapping in the resonator. For the near-zero scattering, the time delays are dramatically enhanced up to the wavepacket duration scale. Similarly, the frequency shift is enhanced to the scale of the spectral width of the pulse. Importantly, the previously known Wigner time-delay formula diverges in the zero-scattering point. Using the weak-measurement theory, we have derived simple non-diverging expressions, which accurately describe the time and frequency shifts in the near-zero scattering regime.

We have observed the theoretically predicted enhanced time delays and their dependences on the parameters in an optical 1D

scattering experiment. We have used Gaussian-like pulses propagating in a nano-fibre with a side-coupled toroidal micro-resonator. The zero transmission coefficient corresponds to the so-called 'critical coupling' known in the theory of resonators. Owing to the high quality of the resonator (narrow linewidth), the duration of the pulses in our experiment was only $\sim 3.5$ times larger than the inverse linewidth. Nonetheless, we clearly observed the predicted resonant behaviour of the time delay, which reached the pulse-duration magnitudes (that is, was amplified by the factor of $\sim 3.5$), both positive (subluminal propagation) and negative (superluminal propagation). Thus, this proof-of-principle experiment provides clear evidence of the described phenomena.

It is important to emphasize that all previously known examples of quantum weak measurements and anomalous wavepacket (or wave-beam) shifts dealt with 2D or 3D systems with internal degrees of freedom (polarization or spin). In sharp contrast to this, we observe similar effects in a 1D scalar wave system. This is possible because of the non-Hermitian nature of this system, which involves complex frequencies and phases, and generates an effectively 2D vortex in the dependence of the scattering coefficient on the complex frequency.

We finally note that the results presented in this work are quite general. They can be applied to any wave system with a near-zero scattering of 1D wave packets. For instance, besides the example considered here, this can be the near-zero reflection from a dissipative cavity in 1D classical-wave systems[31,32,50] or an analogous quantum reflection from a complex double-barrier potential.

**Data availability**. The data that support the findings of this study are available from the corresponding authors upon request.

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

## Acknowledgements

We thank Guanming Zhao (Washington University in St Louis) for providing the microtoroid resonators used in this work. This study was supported by the RIKEN iTHES Project, the MURI Center for Dynamic Magneto-Optics via the AFOSR (grant number FA9550-14-1-0040), Grant-in-Aid for Scientific Research (A), MEXT/JSPS KAKENHI (grants number JP16H01054, JP16H02214, JP15H03704 and JP15KK0164), a grant from the John Templeton Foundation, the Australian Research Council and ARO Grant Number W911NF-16-1-0339.

## Author contributions

M.A., K.Y.B. and Y.P.B. contributed equally to the work. K.Y.B., Y.P.B. and Ş.K.Ö. conceived the idea. K.Y.B., Y.P.B. and A.G.K. developed the theory. A.G.K. contributed the detailed quantum weak-measurement interpretation of the problem. Ş.K.Ö. and T.Y. designed the experiments. M.A. performed the experiments with help from R.I., T.Y. and Ş.K.Ö., Y.P.B. analysyed the experimental data with help from M.A., K.Ş.Ö., K.Y.B. wrote the manuscript with input from all the authors. N.I. and F.N. supervised the research.

## Additional information

**Competing financial interests:** The authors declare no competing financial interests.

