## [Peer Review File · Nature Communications]

REVIEWERS' COMMENTS:

Reviewer #1 (Remarks to the Author):

The authors discuss inelastic resonant scattering of a Gaussian wave packet close to a zero of the complex scattering coefficient and they show, both theoretically and experimentally, that a large time delay and frequency shift occurs.

The results presented by the authors are very interesting, in particular the fact that they found a very elegant and nice explanation of the nature of the time and frequency delays, based on the analogy of this 1D phenomenon with the well known (2D) Goos Hänchen shift. The work is clearly written and very well references, and the result are, in my opinion, very interesting, innovative and have potential for applications.

I therefore think that these results deserve publications in Nature Communications. The only remark I would like to share with the authors is that it would be very nice if they include, in the conclusions for example, a brief discussion whether their findings could have some impact in real applications, such as, for example, optical communications.

Reviewer #2 (Remarks to the Author):

The authors explore the scattering of a wave packet near a zero of the transmission coefficient. They derive a complex time delay whose real and imaginary parts are related to a time shift and a frequency shift. They find an enhancement of the time delay to a value of the order of the pulse duration. Experimental results agree with predictions based on the theory of quantum weak measurements. The results are interesting and the paper is well written. I recommend publication.